# Effect of Mycorrhiza Fungi, Preceding Crops, Mineral and Bio Fertilizers on Maize Intercropping with Cowpea

**Atef A. M. Zen El-Dein [1], Mohamed H. M. Koriem [1], Moodi Saham Alsubeie [2], Reem A. Alsalmi [3], Abdurrahman S. Masrahi [4], Nadi Awad Al-Harbi [5], Salem Mesfir Al-Qahtani [5], Mamdouh M. A. Awad-Allah [6,*] and Yaser A. A. Hefny [1]**

[1] Crop Intensification Research, Field Crops Research Institute, Agricultural Research Center, Giza 12619, Egypt
[2] Biology Department, College of Science, Imam Mohammad Ibn Saud Islamic University (IMSIU), Riyadh 11623, Saudi Arabia
[3] Department of Biology, College of Science, AlBaha University, AlBaha 1988, Saudi Arabia
[4] Biology Department, Faculty of Science, Jazan University, Jazan 45142, Saudi Arabia
[5] Biology Department, University College of Tayma, University of Tabuk, P.O. Box 741, Tabuk 47512, Saudi Arabia
[6] Rice Research Department, Field Crops Research Institute, Agricultural Research Center, Giza 12619, Egypt
* Correspondence: momduhm@yahoo.com

**Abstract:** One filed experiment was carried out to study the effect of Arbuscular Mycorrhiza fungi and three preceding winter crops, i.e., Meskawy cultivar of Egyptian clover berseem (*Trifolium alexandrinum* L.), Careem cultivar of sugar beet (*Beta vulgaris*) and Sakha 94 cultivar of wheat (*Triticum aestivum*) and five fertilizer combinations as treatments of NPK mineral and bio fertilizer which included 100% NPK (T1), 75% NPK + arbuscular mycorrhiza fungi (AMFs) (T2), 50% NPK + arbuscular mycorrhiza fungi (AMFs) (T3), 75% NPK + mycrobein (T4) and 50% NPK + mycrobein (T5) on maize intercropping with cowpea. The results showed that berseem was the best as a preceding crop and gave the highest values of maize and cowpea, followed by sugar beet as a preceding crop. While wheat recorded the lowest values. Fertilizer treatments had significant effect on all maize and cowpea traits. The treatment 75% NPK + arbuscular mycorrhiza fungi (AMFs) (T2) gave the highest values. Meanwhile, no significant differences were found between fertilizer treatments T1 (100% NPK mineral) and T2 (75% NPK + arbuscular mycorrhiza fungi (AMFs)) combination on all studied characters of maize. The interaction had a significant effect on most studied characters of maize and cowpea in the two growing seasons. The cultivation of the two components of intercropping after berseem with T2 fertilizer recorded the highest values. Mixing the third cut of cowpea with maize straw increased significantly the quality and digestibility of forge in both seasons. Planting after berseem and T2 fertilizer gave the highest values as yield advantageous for land equivalent ratio (LER) and relative crowding coefficient (K) which recorded 1.51 and 1.6 and 9.45 and 15.35 in the first and second seasons, respectively. The increases in net return were 3955.67 and 5062.50 L.E., which equates to a percentage of 34.25 and 44.71%, by cultivation intercropping component after berseem and T2 fertilizer treatment (75% NPK + arbuscular mycorrhiza fungi (AMFs)) compared with maize pure stand in first and second seasons, respectively.

**Keywords:** preceding crops; intercropping; arbuscular mycorrhiza; maize; cowpea; fertilization; yield; quality; land use efficiency; net return

## 1. Introduction

Maize (*Zea mays* L.) is the third most important crop in Egypt and the world after rice and wheat, since it is a dual-purpose crop (grain and fodder) and plays an important role in human and animal nutrition [1]. Maize used as forage for livestock feed provides a large amount of energy to the livestock that feed on it, and it is harmless and free of substances such as oxalic acid and prussic acid found in plants such as sorghum. Another advantage

of maize fodder is that it can be consumed by livestock at all stages of growth [1,2]. It is known that maize contributes greatly to enhancing the livelihood and economy in rural areas, and therefore it is grown in a large area annually to produce grain and forage from maize [3]. Cowpea (*Vigna unguiculata* L.) is a multi-cut fodder crop during the growing season. Cowpea is usually climbing growth habit and characterized by its high protein content. It is successfully grown and cultivated in irrigated arable lands in the Mediterranean basin, as well as in low-fertility lands [4]. Forage cowpea as a legume crop contributes to increasing the fertility and balance of soil nutrients, as it has the ability to fix nitrogen from the atmosphere. previous studies found that there was a significant increase in soil organic carbon when cowpea was intercropped with maize, compared to single maize. Moreover, due to the nature of vegetation growth, it was found to be very effective in reducing soil erosion and maintaining soil moisture content [5]. However, the different root structure of maize and cowpea as well as the different soil depth in which the roots are located allow the crops to capture soil moisture at different depths, increasing the water use efficiency of the intercropping system [6]. Based on this, it is clear that intercropping cowpea with maize is effective in maintaining the soil moisture content by about 15.98% during the active period of crop development and by up to 16.70% after crop removal [7,8]. Based on this, the intercropping of maize–cowpea is very useful in countries that suffer from rising water problems, such as Egypt, as this method helps to save an important percentage of irrigation water. However, due to the limited amount of arable land and not enough for the cultivation of major cereal crops, it is not possible to allocate a full area for the production of fodder. However, in this case, using the wide spaces between maize plants to grow promising forage legumes like cowpeas would be a wise decision for farm management [9].

Previous studies proved that intercropping, as a means of agricultural intensification, was one of the vital practices in increasing the efficiency of land use, thus raising the yield advantage and increasing the economic value of the agricultural system. Therefore recently, intercropping has been considered an effective and important strategy to enhance the resilience of the farming system to the risks of climate change [10]. The intercropping between cereals and legumes is a widely proposed strategy for developing a sustainable food and forage production system and to reduce partially overcome food and forage gaps, especially in developing countries with limited agricultural inputs [6,8,11,12]. To practice intercropping properly, each of the component crops needs sufficient surrounding area to adapt to competition between plants, and this is kept to a minimum a reasonable amount of yield is ensured from each crop [13]. Kitonyo et al., 2013 [14] showed that in the maize-legume intercropping system, maize recorded gains due to its rapid initial growth, longer stems and larger root system, all of which leads to its exposure to more limited competition than the accompanying legume crop [15]. In this direction, suggested that suitable spatial arrangements are among the most important factors that enhance the superiority of intercropping between maize-legumes over the single maize crop. Hence, the pattern of intercropping should affect the growth and performance of cowpeas because it determines the optimal space available to them for meet their needs of various growth inputs [12,16].

The quality of preceding crops and intensive land use with the continuous cultivation of similar and closely related crops greatly affects the fertility, health and production capacity of the soil and the growth of crops, causing concerns about its potential to have long-term adverse effects on environmental pollution. Therefore, the agricultural cycle or the sequence and succession of planting crops in the soil affects and is affected by plant growth, soil fertility and the preceding crops. It is necessary to build that system in a correct manner and the existence of a useful crop and often legumes within this sequence [17]. This summarizes our current understanding of the "most well-known" mechanisms of crop rotation, and discusses other mechanisms (such as changes in rhizosphere biology, allelopathic processing, soil structure) that may help to fully account for the benefits of rotation that have been observed by agricultural producers for more than 2000 years.

Succeeding crops, i.e., maize and wheat, had grain yields and components after legumes that were significantly higher than those after non-legumes [18–20]. Moreover, several researchers reported that, the soil macro and micronutrients, organic matter and C/N ratio of residues were affected by the preceding crops and consequently affect yield and yield components of the following crop [21–24].

The increase in population results in increased consumption and demand for food in the world and also the maintenance of food quality. Moreover, the proposed agricultural strategies for increasing and improving food in general must avoid the increase in high input costs. Recently, biofertilizers, including mycrobein and arbuscular mycorrhiza fungi (AMFs), have been receiving increasing interest and appreciation from scientists due to the fact that they do not pose any environmental threats, usually have a long-lasting effect, and if properly managed, bio-fertilizers the same crop can be produced using recommended doses of chemical fertilizers [25]. Arbuscular mycorrhizas fungi receive carbon from plants. Arbuscular mycorrhizas can mobilize different nutrients to the plant and the plant supplies the fungi with mainly organic carbon. Among them, phosphorus is the most important element mobilized [26]. Mycorrhiza biofertilizer produced healthy plants and improved seed quality [27]. In addition, mycorrhizal fungi increase the available phosphorous in the soil [28]. Several studies mentioned that the use of Arbuscular Mycorrhiza and bio-fertilization led to a significant reduction of mineral fertilization and a highly significant increase in growth and yield and a high increase in the quality of maize grain [20].

The purpose of this study is to know the effect of inoculation with bio-fertilizers containing arbuscular mycorrhiza fungi and Mycobrein is a biofertilizer composed by free N-fixing bacteria such as Azotobacter and some P-dissolving bacteria such as Bacillus and Pseudomonas as well as chemical fertilizers on maize and cowpea, to study the extent to which phosphorous can be biologically reformed by changing it from an insoluble form to a soluble form available to maize and cowpea shell. On the other hand, we plan to select and apply the best mixture of bio-fertilizer and part of chemical fertilizer to reduce the used amounts of chemical fertilizer, which reduces chemical fertilizer leaching from the soil and reduces pollution.

The objectives of this study were to determine the effect of inoculation with mycorrhiza and mycrobein strains, preceding crops, intercropping, N, P and K fertilization, as well as their interaction on grain yield, yield forage, and its components as well as chemical composition of maize and cowpea seeds.

## 2. Materials and Methods

### 2.1. Experimental Location and Treatments

One field experiment was carried out in Etay El-Baroud Experimental Station at El-Beheira Governorate, Agricultural Research Center, Egypt through 2019/2020 and 2020/2021 seasons to study the effect of three preceding crops and five NPK mineral fertilizers and biofertilizers treatments on yield and yield components of maize (*Zea mays*, L.) cultivar (Y T C, 368) and cowpea cultivar (Kafr El-Shekh) intercropping as follow:

A.   Preceding crops:

    1.   Berseem (Meskawy, cv.).
    2.   Sugar beet (Careem, cv.).
    3.   Wheat (Sakha 94, cv.).

B.   five treatments of NPK mineral fertilizer and bio fertilizers:

    T1: (100% mineral NPK).
    T2: (75% mineral NPK + Arbuscular Mycorrhiza fungi).
    T3: (50% mineral NPK + Arbuscular Mycorrhiza fungi).
    T4: (75% mineral NPK + Mycrobein).
    T5: (50% mineral NPK + Mycrobein).

In this case, 100% maize + 50% cowpea intercropping pattern was used for all intercropping treatments. Furthermore, maize and cowpea purely stand as recommended

for each crop. All other cultivation treatments were carried out in accordance with the recommendations of the Ministry of Agriculture and Land Reclamation

### 2.2. The Fertilization Rates

For mineral fertilizers, the rates 100%, 75% and 50% of phosphorus equal 150 kg, 112.50 kg and 75 kg P2O5, respectively, for 100% maize and 50% cowpea together were added when preparing the land for cultivation. In this case, 100% N (120 + 22.50 kg N for 100% maize + 50% cowpea), 75% N (90 + 16.88 kg N for 100% maize + 50% cowpea) and 50% N (60 + 11.25 kg N for 100% maize + 50% cowpea). The rates 100%, 75% and 50% of nitrogen equal 311.83 kg, 233.87 kg and 155.92 kg urea (46.50% N), respectively, for 100% maize + 50% cowpea were added in two equal doses before the first and second irrigation. The rates 100%, 75% and 50% of potassium equal 75 kg, 56.25 kg and 37.50 kg $P_2O_5$, respectively, for both 100% maize and 50% cowpea together were added before the first irrigation. Whereas biofertilizers were applied to maize grains or cowpea seeds at the rate of 800 g/fed of Mycorrhiza or Microbean before sowing directly, then planting and irrigation carried out immediately. Mycobrein is a biofertilizer composed by free N-fixing bacteria such as Azotobacter and some P-dissolving bacteria such as Bacillus and Pseudomonas.

### 2.3. Experimental Design

The experimental design was a split-plot with three replications, preceding winter crops were located in the main plot, while fertilizer treatments were randomized distributed to the sub-plots. The area of each sub-plot was 4 ridges (70 cm width), and the length of the ridge was 3.50 m (plot area was 9.80 $m^2$ = 1/428.57 of feddan).

The soil analyses of the experimental site before sowing components intercropping and after harvesting of preceding crops, Table 1.

**Table 1.** The physical and chemical analysis of experimental soil before sowing of preceding crops during 2020 and 2021 seasons.

| Soil Properties | Soil Texture | Sand% | Silt% | Clay% | PH | Organic Matter% | Available N (%) | Available P (%) | Available K (%) | EC (m mhos) cm$^{-1}$ (1;5) |
|---|---|---|---|---|---|---|---|---|---|---|
| 2019/20 | Clay | 7.08 | 32.53 | 60.39 | 7.71 | 2.10 | 0.017 | 0.010 | 0.221 | 1.5 |
| 2020/21 | Clay | 7.09 | 32.96 | 59.95 | 7.79 | 2.14 | 0.017 | 0.011 | 0.301 | 1.6 |

The soil samples of the experimental sites were taken at the depth of 0–30 cm, to determine the physical and chemical analysis of the soil after sowing of preceding crops as shown in Table 2. According to the methods described by Page et al. (1982) [29], the soil characteristics were determined from a soil extract of 1:1 used for measuring soil PH using PH meter and potassium by flame photometer instrument. The total nitrogen was measured by kjeldhal method using a micro-Kjeldahl instrument [30]. Organic matter was measured in the soil by wet digestion with concentrated sulfuric acid using method Black et al. (1965) [31], Tables 1 and 2.

**Table 2.** Some physical and chemical properties of the soil of the experiment site after harvesting preceding crops.

| Soil Variable | Berseem | | Sugar Beet | | Wheat | |
|---|---|---|---|---|---|---|
| | 2020 | 2021 | 2020 | 2021 | 2020 | 2021 |
| PH | 8.11 | 7.99 | 7.93 | 7.85 | 8.03 | 7.89 |
| Organic matter (%) | 3.05 | 301 | 2.29 | 2.21 | 2.09 | 2.07 |
| Available N (%) | 0.075 | 0.072 | 0.061 | 0.059 | 0.039 | 0.040 |
| Available P (%) | 0.0277 | 0.0275 | 0.0267 | 0.0236 | 0.021 | 0.0206 |
| Available K (%) | 0.0684 | 0.0697 | 0.0691 | 0.0681 | 0.0545 | 0.0551 |

### 2.4. Planting Date

The preceding crops were planted on November 15th and 17th, and harvested on the first of May in both seasons, while maize and cowpea were planted on June 1st and 2nd and harvested on October 1st and 3rd in the first and second seasons, respectively. Weather conditions during the 2020 and 2021 planting seasons are shown in Table 3.

**Table 3.** The weather conditions during the 2020 and 2021 growing seasons.

| Month | Temperature (°C) | | | | Relative Humidity (%) | |
|---|---|---|---|---|---|---|
| | 2020 | | 2021 | | 2020 | 2021 |
| | Min | Max | Min | Max | | |
| May | 17 | 32 | 18 | 34 | 80 | 79 |
| June | 21 | 34 | 22 | 35 | 81 | 80 |
| July | 24 | 35 | 23 | 36 | 85 | 83 |
| August | 24 | 36 | 23 | 37 | 75 | 74 |
| September | 23 | 32 | 23 | 32 | 70 | 71 |
| October | 18 | 31 | 17 | 30 | 69 | 70 |

Meteorological records of Central Laboratory for Agriculture Climate (Source: Etay El-Baroud Research Station) El-Beheira Governorate of the Agriculture Research Center, Egypt, 2020 and 2021.

### 2.5. The Method Intercropping

Maize and cowpea were planted in a two ridges maize to two ridges cowpea in the summer season after the winter crops. Maize was thinned to two plants/hill with a distance of 30 cm between hills on one side of the ridge, while cowpea was planted in hills and two plants/hill with a distance of 15 cm between the hills on both sides of the ridge (100% corn + 50% cowpea).

#### 2.5.1. Maize Measurements

Harvest times at the age of 120 days from sowing maize samples of ten plants were chosen randomly from each sub plot to estimate plant height (cm), number of grain/rows, 100-grain weight (g), and grain weight/ear (g). The maize was then harvested by hand by cutting the stalks just above ground level. The biological yield (ears + straw by ton/fed) per sub-plot was weighed immediately in the field. After that, ears per each sub-plot were separated and weighed to determine ear yield (ton/fed), then shelled to determine grain yield (ton/fed). The maize straw was determined, while crude protein percentage (CP%): nitrogen (N) content was analyzed using the Kjeldahl procedure N × 6.25 (AOAC, 2012) [32], crude fiber percentage (CF%) and crude ash percentage (CA%). Next, the maize straw was mixed with the 3rd third cut of cowpea to raise the value of the fodder.

#### 2.5.2. Cowpea Measurements

Cowpea was harvested on three cuts, the first cut was taken at 55 days of sowing, and the second cut was taken at 40 days, after the first cut and third cut was taken at 30 days after the second cut, which was weighed then mixed with maize straw. The dried samples were ground to 1 mm particle size for feed quality analysis. The nitrogen (N) content was analyzed using the Kjeldahl procedure (AOAC, 2012) [32], then the crude protein percentage (CP%) was calculated as N × 6.25. The crude fiber percentage (CF%) was determined using the Soxhlet procedure (AOAC, 2012) [32]. The crude ash content percentage (CA%) was analyzed and determined by burning the samples in a muffle oven at 550 °C for 3 h (AOAC, 2012) [32].

#### 2.5.3. Competitive Relationships

When the values of LER and K were greater than1, there was a yield advantage, when LER and K were equal to 1, there was no yield advantage, and, when it was less than 1, there was a disadvantage [33].

1- Land equivalent ratio (LUR) and Yield Advantage: It was measured and calculated as the sum of the fractional yield (t feda$^{-1}$) of maize and cowpea crops relative to their individual crop yield [34] as following:

$$LER = Y_{ab}/Y_{aa} + Y_{ba}/Y_{bb}$$

where: $Y_{ab}$ is yield of corn ‹a‹ intercropped with cowpea ‹b‹, $Y_{aa}$ is pure stand yield of corn ‹a‹, $Y_{ba}$ is yield of forage cowpea ‹b‹ intercropped with maize ‹a‹, and $Y_{bb}$ is pure stand yield of cowpea ‹b‹.

Land equivalent ratio was calculated twice, i.e., using both the 50% and 100% pure cowpea stands.

2- Relative crowding coefficient (RCC or K): was determined as the measure of the relative dominance of one species over the other in a mixture [35]. The K was calculated as follows:

$$K = (K_a \times K_b), \text{ where } K_a = Y_{ab} \times Z_{ba}/((Y_{aa} - Y_{ab}) \times Z_{ab}), \text{ and } K_b = Y_{ba} \times Z_{ab}/((Y_{bb} - Y_{ba}) \times Z_{ba})$$

where $Z_{ab}$ and $Z_{ba}$ were the proportions of cereal and legume in the mixture, respectively. When the values of LER and K were greater than 1, there was a yield advantage, when LER and K were equal to 1, there was no yield advantage, and, when it was less than 1, there was a disadvantage [33].

3- Aggressivity (A), which is often used to determine the competitive relationship between two crops used in mixed cropping [36]. The aggressivity was formulated as follows:

$Aa = (Y_{ab}/Y_{aa} \times Z_{ab}) - (Y_{ba}/Y_{bb} \times Z_{ba})$
$Ab = (Y_{ba}/Y_{bb} \times Z_{ba}) - (Y_{ab}/Y_{aa} \times Z_{ab})$, [35].

For example, if Aa = 0, both crops are equally competitive, if Aa is positive, then the a species is dominant, if Aa is negative, then the a is weak.

### 2.5.4. Economic Evaluation

Net return = Gross return − production costs, the gross income for each crop was calculated in the Egyptian pounds per feddan. In this case, 5400 L.E. per ton of grain (The price of yellow corn in Egypt, 2021), maize straw was ton calculated by 100 L.E. per ton [37], with 450 E.L. per ton of green fodder cowpea, according to (Alfallahalyoum, feed prices, 2021) [38].

### *2.6. Statistical Analysis*

The data obtained were analyzed by split-plot design according to Snedecor and Cochran [39]. The treatment's means were compared by using the least significant differences (L.S.D.) at 5% of probability and Duncan's Multiple Range Test which described by Duncan 1955 [40]. The analysis of variance (ANOVA) was computed using CoStat V 6.4 (2005) program [41].

### **3. Results and Discussion**
### *3.1. Maize*
### 3.1.1. Effect of Preceding Crops

The data in Table 4 showed that all studied characters were significantly affected by preceding crops in 2020 and 2021 seasons. The maximum values of the maize yield and its components were observed for growing after berseem. The highest values were 2.34 and 2.51 ton/fed for grain yield, 6.58 and 6.62 ton/fed for straw yield and 8.92 and 9.13 ton/fed for biological yield were obtained when grown maize after berseem, followed by grown maize after sugar beet in both seasons, while the lowest values 2.03 and 2.08 ton/fed for grain yield, 5.53 and 5.66 ton/fed for straw yield and 7.56 and 7.79 ton/fed for biological yield were recorded when grown maize after wheat in both seasons, respectively, Figure 1. Similar results were obtained by Veneklaas et al. [42]. These findings

resulted from cultivation after berseem (Egyptian clover) can be traced back to the mobilization of soil and fertilizer N through the exudation of organic acid anions such as citrate, maltase and other compounds from their roots in addition to N fixing from root nodes. This method enables some of these species to obtain nitrogen, phosphorous and potassium from soil sources that are not readily available to uncultivated crops grown with or alternating crops. According to Meek et al. [43], a crop rotation following Egyptian clover (berseem) with maize or a crop that has the same nitrogen uptake pattern instead of beans will provide nitrogen fertilizer, reduce $NO_3$ levels in the soil, and reduce the potential for $NO_3$ leaching. $NO_3$-N less deep leaching associated with wheat lentil rotation due to better synchronization of nitrogen uptake from decomposition of lentil residues compared to continuous fertilized wheat [44]. Grain yield for maize under intercropping with cowpea was more than 70% of its monoculture crop in both seasons, although maize intercropping is sowing only in 50% of the single maize space. These results are due to intercropping are intensification crop production and exploiting more efficient environments with limiting or potentially limiting growth resources, these findings were in agreement with those obtained by [36,45].

**Table 4.** The effect of preceding crops on maize yield and some of its components under intercropping system compared with sole maize during 2020 and 2021 seasons.

| | Preceding Crops | Plant Height (cm) | | No. of Grain /Row | | 100-Grain Weight (g) | | Grain Weight/Ear (g) | | Grain Yield (Ton /Fed) | | Straw Yield (Ton/Fed) | | Biological Yield (Ton/Fed) | |
|---|---|---|---|---|---|---|---|---|---|---|---|---|---|---|---|
| | | 2020 | 2021 | 2020 | 2021 | 2020 | 2021 | 2020 | 2021 | 2020 | 2021 | 2020 | 2021 | 2020 | 2021 |
| | Berseem | 237.29 | 234.07 | 29.94 | 30.81 | 29.36 | 30.09 | 102.04 | 108.62 | 2.34 | 2.51 | 6.58 | 6.62 | 8.92 | 9.13 |
| | Sugar beet | 227.89 | 227.55 | 26.40 | 29.83 | 27.58 | 28.09 | 97.19 | 102.55 | 2.27 | 2.36 | 6.47 | 6.45 | 8.70 | 8.80 |
| | Wheat | 199.13 | 202.30 | 20.84 | 23.93 | 27.25 | 28.11 | 87.80 | 90.61 | 2.03 | 2.08 | 5.53 | 5.66 | 7.56 | 7.79 |
| | L.S.D.at 5% | 3.68 | 3.18 | 0.28 | 0.30 | 0.63 | 0.74 | 1.40 | 1.03 | 0.20 | 0.11 | 0.07 | 0.16 | 0.13 | 0.10 |
| Sole maize | After berseem | 252.16 | 249.24 | 31.79 | 30.91 | 28.91 | 29.52 | 125.03 | 122.05 | 3.25 | 3.18 | 8.11 | 8.02 | 11.26 | 11.13 |
| | After S. beet | 243.07 | 239.87 | 31.53 | 30.87 | 28.65 | 29.31 | 123.73 | 120.23 | 3.18 | 3.11 | 8.09 | 7.97 | 11.18 | 11.05 |
| | After wheat | 231.25 | 229.61 | 30.55 | 29.75 | 27.66 | 28.50 | 115.11 | 112.13 | 2.93 | 2.95 | 7.78 | 7.67 | 10.90 | 10.72 |
| | Average | 242.16 | 239.57 | 31.29 | 30.51 | 28.41 | 29.11 | 120.29 | 117.47 | 3.12 | 3.08 | 8.00 | 7.89 | 11.12 | 10.97 |

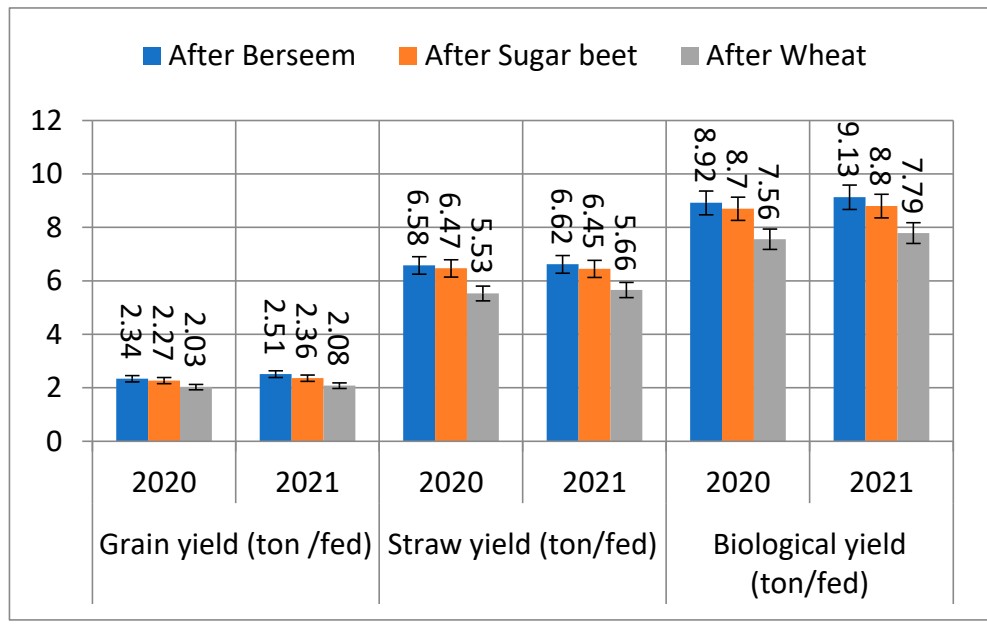

**Figure 1.** The effect of preceding crops on grain yield (ton/fed), straw yield (ton/fed) and biological yield (ton/fed) of maize during 2020 and 2021 seasons.

### 3.1.2. Effect of NPK and Biofertilizers

All studied characters were significantly affected by fertilizer treatments as presented in Table 5. In this case, 75% NPK + mycorrhiza (T2) resulted the highest values in all studied characters which estimated by 2.29 and 2.43 ton/fed for grain yield, 6.40 and 6.46 ton/fed for straw yield and 8.69 and 8.87 ton/fed for biological yield in the first and second seasons, respectively. Treatments for two (75% NPK + mycorrhiza) (T2) and 100% NPK (T1), followed by 75% NPK + mycrobein (T4) treatment enhanced the highest values in both seasons, with no significant difference between T1 and T2 in both seasons. Meanwhile, 50% NPK + mycorobein (T5) recorded the lowest values in all studied characters, with values 2.03 and 2.11 ton/fed for grain yield, 5.92 and 6.05 ton/fed for straw yield and 7.95 and 8.00 ton/fed for biological yield in the first and second seasons, respectively, as per Figure 2. These results may be due to the fact that 75% NPK + mycorrhiza increases the phosphorus available and optimal fertilizer dose. Mycorrhiza led to a continuous supply of P as well as few little quantitative of N and K to plants under intercropping condition, in addition to the fixed nitrogen and other benefits resulting from the legume plant and the different layers of growth of cowpea and maize roots, which leads to an increase in the metabolism, photosynthesis and cultivation of plants, which leads to an increase in the yield and its components [26]. Mycorrhiza fungi increase available phosphorous, fertilizers derived from animals, plants, and other types of waste that include microorganisms that act in relation to fixing nitrogen, phosphorous and other nutrients in soil, and biofertilizers increase soil properties [46]. The highest percentage of phosphorous, grain protein, grain length, grain width, fresh forage yield, and raw curd were obtained with the vermicompost containing 60 kg N/h + (farmyard manure + 3 ton chickpea waste/h) + phosphate-solubilizing bacteria treatments [47]. The combined use of a mixture of biofertilizer (AMF, Bacillus Circulans and Azotobacter chrocoocum) with organic fertilizers enhanced maize growth, yield and nutrient uptake. Furthermore, bio-organic fertilization improved the soluble sugars, starch, carbohydrate, protein and amino acid contents of the corn seed. Moreover, the bio-organic fertilization causes a significant increase in microbial activity by enhancing the enzymes of acid phosphatase and dehydrogenase, number of bacteria, and fungal colonization levels in maize roots compared to chemical fertilization. In addition, bio-organic fertilizers improved α-amylase and gibberellins activities and transcript levels, as well as reduced the level of abscisic acid in seeds compared to chemical fertilizers [48]. Based on the results obtained from bio-organic fertilization on growth parameters and maize yield, he recommended using it as an alternative tool to reduce chemical fertilizers.

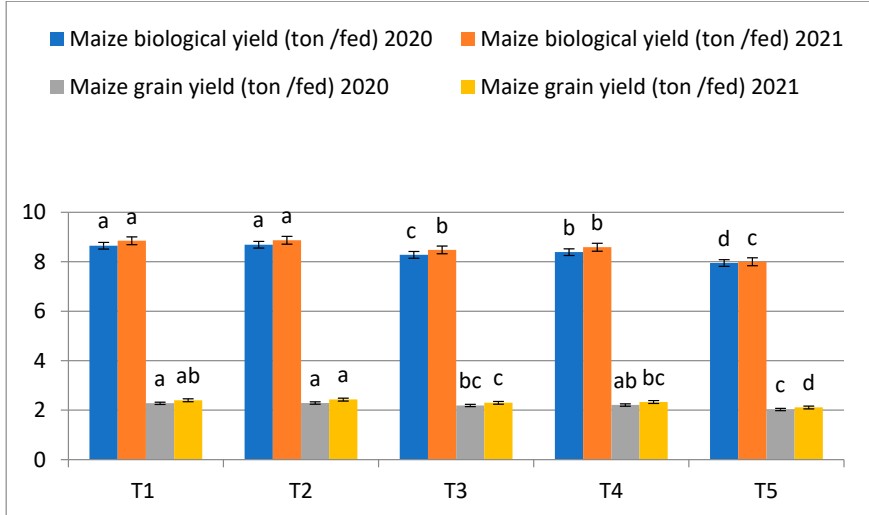

**Figure 2.** The effect of fertilizer treatments on maize biological and grain yield (ton/fed) under intercropping system during 2020 and 2021 seasons. Different alphabetic letters represent the significant differences among the treatments at $p < 0.05$, according to Duncan's test.

**Table 5.** The effect of fertilizer treatments on maize yield and some of its components under intercropping system during 2020 and 2021 seasons.

| Fertilizer Treatment | Plant Height (cm) | | No. of Grain/Row | | 100-Grain Weight (g) | | Grain Weight/Ear (g) | | Grain Yield (Ton/Fed) | | Straw Yield (Ton/Fed) | | Biological Yield (Ton/Fed) | |
|---|---|---|---|---|---|---|---|---|---|---|---|---|---|---|
| | 2020 | 2021 | 2020 | 2021 | 2020 | 2021 | 2020 | 2021 | 2020 | 2021 | 2020 | 2021 | 2020 | 2021 |
| T1 (100% NPK) | 224.13 | 224.04 | 27.23 | 29.11 | 29.20 | 29.72 | 99.15 | 104.71 | 2.28 | 2.40 | 6.38 | 6.44 | 8.65 | 8.85 |
| T2 (75%NPK+AMF) | 224.37 | 225.11 | 27.53 | 29.21 | 29.68 | 30.60 | 99.17 | 104.93 | 2.29 | 2.43 | 6.40 | 6.46 | 8.69 | 8.87 |
| T3 (50%NPK +AMF) | 223.18 | 222.62 | 25.22 | 28.08 | 27.98 | 28.61 | 95.12 | 100.05 | 2.19 | 2.30 | 6.10 | 6.16 | 8.28 | 8.48 |
| T4 (75%NPK +Mycrobein) | 221.00 | 219.17 | 25.39 | 28.42 | 28.37 | 29.19 | 96.83 | 101.36 | 2.21a | 2.33 | 6.18 | 6.24 | 8.39 | 8.59 |
| T5 (50%NPK +Mycrobein) | 214.52 | 215.59 | 23.26 | 26.14 | 25.09 | 25.70 | 88.11 | 91.92 | 2.03 | 2.11 | 5.92 | 6.05 | 7.95 | 8.00 |
| L.S.D.at 5% | 8.24 | 4.84 | 0.49 | 0.28 | 0.67 | 0.68 | 1.38 | 2.43 | 0.17 | 0.10 | 0.15 | 0.13 | 0.19 | 0.11 |
| Interaction | ns | ns | * | * | ns | ns | * | * | ns | * | * | ns | * | * |

$* = p < 0.05$, L.S.D.: least significant differences at 5% of probability, ns: non-significant differences, T1: 100% mineral NPK, T2: 75% mineral NPK + Arbuscular Mycorrhiza fungi, T3: 50% mineral NPK + Arbuscular Mycorrhiza fungi, T4: 75% mineral NPK + Mycrobein, T5: 50% mineral NPK + Mycrobein.

### 3.1.3. The Interaction Effects

The interaction between preceding crops and fertilizer treatments had significant effect on number of grains/row, grains weight/ear and biological yield ton/fed in both seasons as shown in Table 6. The grain yield ton/fed was significantly affected by the interaction in the second season, while straw yield ton/fed influenced by the interaction in the first season. Maize grown after berseem and fertilized by 75%. NPK + Mycorrhiza (T2) gave the highest values of grain yield, straw yield and biological yield which had an estimated 2.62 ton/fed for grain yield in the second season and 6.77 ton/fed for straw yield in the first season, while 9.23 and 9.53 ton/fed for biological yield in both seasons. However, maize grown after wheat and fertilized by 50% NPK + Mycorobein (T5) recorded the lowest values in most studied treats, where there was 1.97 ton/fed for grain yield in the second season and 5.29 ton/fed for straw yield in the first season, while 7.21 and 7.35 ton/fed for biological yield in the two growing seasons. Meanwhile no differ significances found between 75% NPK + mycorobein (T4) and 100% NPK (T1) with grown maize after berseem under intercropping condition for grain weight/ear, grain yield, straw yield and biological yield per fed. These results may be due to planting maize after berseem and fertilized by biofertilizer lead to increase physical and chemical properties soil compared to NPK fertilizer alone. These results were in harmony with that obtained by Mahrous et al. [49]. Increased populations of arbuscular mycorrhizal fungi (AMF) in soils after cultivation of mycorrhizal host crops improved AM colonization of succeeding crops in the following season, which consequently enhanced P uptake, early growth and yield of succeeding maize [50].

### 3.2. Cowpea
### 3.2.1. Effect of Preceding Crops

The preceding crops had significant effect on 1st cut, 2nd cut, 3rd cut and total yield in the two seasons as shown in Table 7. The grown cowpea after berseem achieved the highest values were 8.44 and 8.88 ton/fed for 1st cut, 3.86 and 3.75 ton/fed for 2nd cut, 2.19 and 2.20 ton/fed for 3rd cut as well as 14.49 and 14.83 ton/fed for total yield in the first and second seasons, respectively. While the lowest values 6.73 and 7.10 ton/fed for 1st cut, 2.50 and 2.38 ton/fed for 2nd cut, 1.85 and 1.86 ton/fed for 3rd cut as well as 11.08 and 11.34 ton/fed for total yield were recorded when cowpea grown after wheat for these characters in both seasons, respectively. These results may be due to increase the organic nitrogen produced by the root nodes of berseem. The forge yield/fed for cowpea under intercropping with maize was more than 70% of its monoculture crop in both seasons, although cowpea intercropping is sowing only in 50% of the single cowpea area. Crop inferences with more than one crop are usually built by elements of the crop sequence with beneficial yield, exploiting crop, plant growth status and soil fertility [17].

**Table 6.** The effect of interaction on maize yield and some of its components under intercropping system during 2019 and 2020 seasons.

| | Fertilizer Treatment | No. of Grain/Row | | 100-Grain Weight(g) | | Grain Weight/Ear (g) | | Grain Yield (Ton/Fed) | | Straw Yield (Ton/Fed) | | Biological Yield (Ton/Fed) | |
|---|---|---|---|---|---|---|---|---|---|---|---|---|---|
| | | 2020 | 2021 | 2020 | 2021 | 2020 | 2021 | 2020 | 2021 | 2020 | 2021 | 2020 | 2021 |
| Berssem | T1 | 31.40 | 32.13 | 30.47 | 30.90 | 104.15 | 112.56 | 2.40 | 2.60 | 6.68 | 6.65 | 9.08 | 9.29 |
| | T2 | 31.93 | 32.22 | 30.65 | 31.90 | 105.01 | 113.30 | 2.46 | 2.62 | 6.77 | 6.94 | 9.23 | 9.53 |
| | T3 | 29.80 | 30.73 | 29.85 | 30.43 | 103.99 | 110.04 | 2.32 | 2.53 | 6.61 | 6.57 | 8.93 | 9.18 |
| | T4 | 29.86 | 31.10 | 30.01 | 31.26 | 104.37 | 111.79 | 2.39 | 2.59 | 6.56 | 6.68 | 8.95 | 9.31 |
| | T5 | 26.70 | 27.87 | 25.80 | 25.97 | 92.70 | 95.42 | 2.13 | 2.19 | 6.27 | 6.17 | 8.40 | 8.36 |
| S. beet | T1 | 28.30 | 31.00 | 28.65 | 28.87 | 103.31 | 109.45 | 2.37 | 2.51 | 6.70 | 6.73 | 9.04 | 9.24 |
| | T2 | 28.33 | 30.80 | 28.97 | 30.10 | 101.83 | 108.64 | 2.34 | 2.50 | 6.72 | 6.59 | 9.06 | 9.09 |
| | T3 | 25.80 | 29.80 | 27.20 | 27.70 | 95.88 | 99.47 | 2.20 | 2.29 | 6.35 | 6.45 | 8.55 | 8.74 |
| | T4 | 26.10 | 30.30 | 27.90 | 28.20 | 96.80 | 100.74 | 2.19 | 2.33 | 6.43 | 6.36 | 8.62 | 8.66 |
| | T5 | 23.47 | 27.26 | 25.20 | 25.60 | 88.13 | 94.44 | 2.03 | 2.17 | 6.21 | 6.11 | 8.24 | 8.28 |
| Wheat | T1 | 22.00 | 24.20 | 28.47 | 29.40 | 90.01 | 92.11 | 2.07 | 2.12 | 5.76 | 5.91 | 7.83 | 8.03 |
| | T2 | 22.33 | 24.60 | 29.43 | 29.80 | 90.67 | 92.84 | 2.08 | 2.13 | 5.71 | 5.85 | 7.79 | 7.98 |
| | T3 | 20.07 | 23.70 | 26.90 | 27.70 | 85.50 | 90.63 | 2.04 | 2.08 | 5.33 | 5.45 | 7.37 | 7.53 |
| | T4 | 20.20 | 23.87 | 27.20 | 28.10 | 89.33 | 91.56 | 2.06 | 2.10 | 5.55 | 5.69 | 7.61 | 7.79 |
| | T5 | 19.60 | 23.30 | 24.27 | 25.53 | 83.50 | 85.89 | 1.92 | 1.97 | 5.29 | 5.38 | 7.21 | 7.35 |
| L.S.D.at 5% | | 0.71 | 0.41 | ns | ns | 1.97 | 3.49 | ns | 0.08 | 0.21 | ns | 0.23 | 0.26 |

L.S.D.: least significant differences at 5% of probability, ns: non-significant differences, T1: 100% mineral NPK, T2: 75% mineral NPK + Arbuscular Mycorrhiza fungi, T3: 50% mineral NPK + Arbuscular Mycorrhiza fungi, T4: 75% mineral NPK + Mycrobein, T5: 50% mineral NPK + Mycrobein.

**Table 7.** The effect of preceding crops and fertilizations on cowpea forage cuts during 2020 and 2021 seasons under intercropping with compared with mono cropping of cowpea.

| | Treatments | 2020 | | | | 2021 | | | |
|---|---|---|---|---|---|---|---|---|---|
| | | 1st Cut (Ton /Fed) | 2nd Cut (Ton /Fed) | 3rd Cut (Ton /Fed) | Total Yield (Ton/Fed) | 1st Cut (Ton /Fed) | 2nd Cut (Ton /Fed) | 3rd Cut (Ton /Fed) | Total Yield (Ton/Fed) |
| | Berseem | 8.44 | 3.86 | 2.19 | 14.49 | 8.88 | 3.75 | 2.20 | 14.83 |
| | Sugar beet | 7.67 | 3.26 | 1.98 | 12.91 | 8.12 | 3.01 | 2.00 | 13.13 |
| | Wheat | 6.73 | 2.50 | 1.85 | 11.08 | 7.10 | 2.38 | 1.86 | 11.34 |
| | L.S.D. 5% | 0.115 | 0.068 | 0.025 | 0.143 | 0.372 | 0.134 | 0.014 | 0.389 |
| | F1(100% NPK) | 7.87 | 3.40 | 2.16 | 13.43 | 8.42 | 3.22 | 2.19 | 13.82 |
| | F2(75%NPK+AMF) | 8.00 | 3.59 | 2.20 | 13.79 | 8.55 | 3.41 | 2.21 | 14.17 |
| | F3(50%NPK +AMF) | 7.55 | 3.11 | 1.89 | 12.55 | 7.84 | 2.98 | 1.92 | 12.75 |
| | F4(75%NPK +Mycrobein) | 7.60 | 3.17 | 1.95 | 12.72 | 7.93 | 3.04 | 1.93 | 12.90 |
| | F5(50%NPK +Mycrobein) | 7.05 | 2.74 | 1.84 | 11.63 | 7.43 | 2.59 | 1.82 | 11.84 |
| | L.S.D. 5% | 0.164 | 0.119 | 0.032 | 0.211 | 0.197 | 0.111 | 0.034 | 0.205 |
| | Interaction | * | * | ns | * | * | * | * | * |
| Sole cowpea | After berseem | 10.26 | 6.82 | 3.87 | 20.95 | 10.34 | 6.90 | 3.78 | 21.02 |
| | After S. beet | 9.95 | 6.51 | 3.56 | 20.02 | 10.23 | 6.78 | 3.67 | 20.68 |
| | After wheat | 9.70 | 6.26 | 3.31 | 19.27 | 9.95 | 6.51 | 3.40 | 19.86 |
| | Average | 9.97 | 6.53 | 3.58 | 20.08 | 10.17 | 6.74 | 3.61 | 20.52 |

* = $p < 0.05$, L.S.D.: least significant differences at 5% of probability, ns: non-significant differences, T1: 100% mineral NPK, T2: 75% mineral NPK + Arbuscular Mycorrhiza fungi, T3: 50% mineral NPK + Arbuscular Mycorrhiza fungi, T4: 75% mineral NPK + Mycrobein, T5: 50% mineral NPK + Mycrobein.

### 3.2.2. Effect of NPK and Biofertilizers

The data in Table 7 and Figure 3 revealed that fertilizer treatments were highly significant effects on cowpea cuts in both seasons, while 3rd was significantly affected by interaction in the second season only. The application of 75% NPK + Mycorrhiza (T2) resulted in the highest values, those being 8.00 and 8.55 ton/fed for 1st cut, 3.59 and 3.41 ton/fed for 2nd cut, 2.20 and 2.21 ton/fed for 3rd cut, as well as 13.79 and 14.17 ton/fed

for total yield in the first and second seasons, respectively, followed by 100% NPK (T1). These results may be due to biofertilizer, the plant extends the nutrients slowly throughout the growth seasons and delay aging. In the past period, scientists have increasingly become interested in bio-fertilizers such as rhizobia and Arbuscular Mycorrhiza Fungi. These results are due to the fact that leguminous crops need phosphorous in the formation of energy-rich compounds such as adenosine tri-phosphate (ATP) via photophosphorylation, in other photosynthesis reactions, and the light energy in the presence of chlorophyll is used in the hydrolysis (photolysis of water) in the chloroplasts to oxygen and hydroxyl and the formation of a reducing force represented in the molecules of nicotine amide dinucleotide phosphate (Nicotineamide adenine dinucleotide phosphate ($NADPH_2$). In order to convert carbohydrates into protein within the plant, Mycorrhiza reduces nitrogen use and increases phosphorous available in the soil [28]. This is because of its impact on reducing the quantities of mineral fertilization and, thus, reducing pollution and improving quality [25]. Similar results are obtained by Baghdadi et al. [51] when used for bio-fertilization (BF) found that the application of 50% NPK + 50% chicken manure + BF produced fresh yields of forage and dry matter that were comparable to those produced in 100% nitrogen (N), phosphorous (P), and potassium (K).

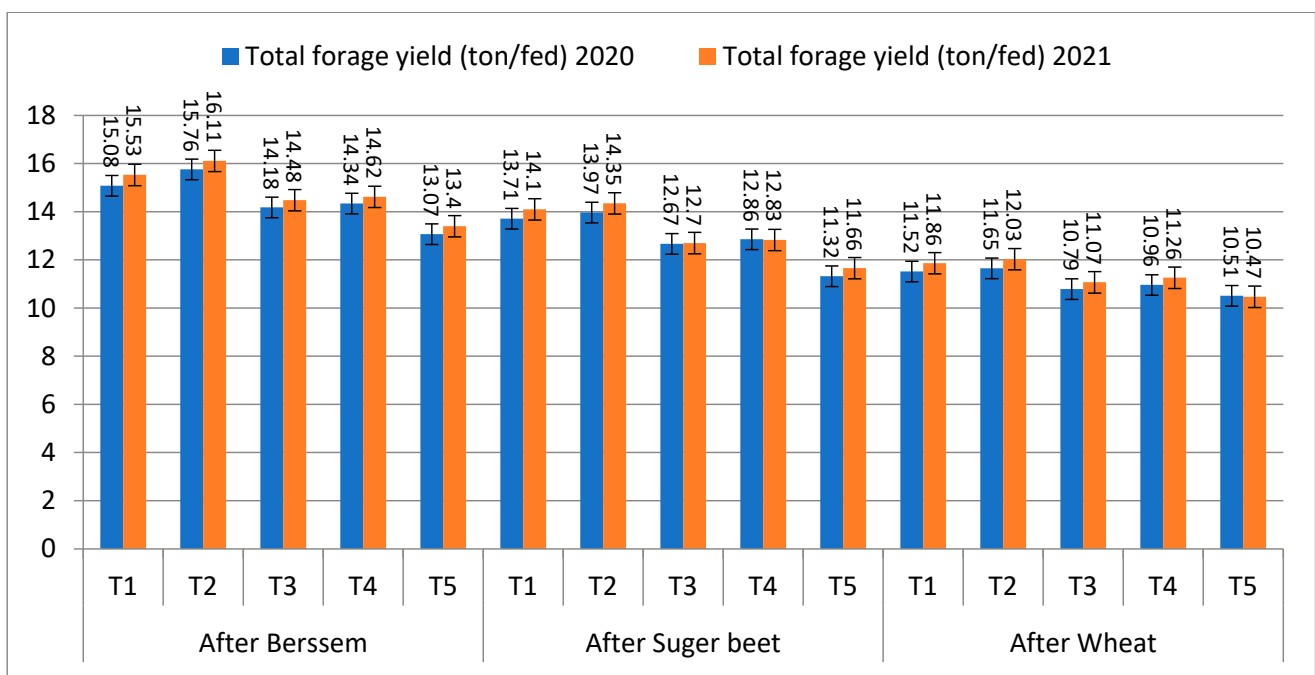

**Figure 3.** The effect of interaction on cowpea forage yield cuts (ton/fed) under intercropping system during 2020 and 2021 seasons. T1: 100% mineral NPK, T2: 75% mineral NPK + Arbuscular Mycorrhiza fungi, T3: 50% mineral NPK + Arbuscular Mycorrhiza fungi, T4: 75% mineral NPK + Mycrobein, T5: 50% mineral NPK + Mycrobein.

3.2.3. The Interaction Effects

The data in Table 8 and Figure 3 indicated that 1st, 2nd cuts and total yield (ton/fed) were significantly affected by interaction in both seasons, while 3rd cut was significantly affected by interaction in the second season only. The highest values of 9.02 and 9.59 ton/fed for 1st cut, 4.33 and 4.13 ton/fed for 2nd cut, 2.41 and 2.39 ton/fed for 3rd cut as well as 15.76 and 16.11 ton/fed for total yield were obtained when fertilized by T2 (75% NPK + Mycorrhiza) in the first and second seasons, respectively. On the contrary, the lowest values were recorded when grown cowpea after wheat and fertilized by of 50% NPK + Mycrobein (T5) with the values 6.63 and 6.75 ton/fed for 1st cutting, 2.17 and 2.04 ton/fed for 2nd cutting, 1.71 and 1.68 ton/fed for 3rd cutting, as well as 10.51

and 10.47 ton/fed for total yield in the first and second seasons, respectively, Table 8 and Figure 3. These results may be due to grown cowpea after berseem, and application of mycorrhiza or mycrobain leads to increased soil properties allowing for the roots to easily penetrate the soil and grow deeper compared with single NPK fertilizer. Intercropping cowpea with maize improved the yield of the crop and the efficiency of nitrogen use efficiency of intercropped maize after berseem cutting [52].

**Table 8.** The effect of interaction on cowpea forage yield cuts (ton/fed) under intercropping system during 2020 and 2021 seasons.

|  | | 2020 | | | | 2021 | | | |
|---|---|---|---|---|---|---|---|---|---|
|  | Treatments | 1st Cut (Ton /Fed) | 2nd Cut(Ton /Fed) | 3rd Cut (Ton /Fed) | Total (Ton /fed) | 1st Cut (Ton /Fed) | 2nd Cut(Ton /Fed) | 3rd Cut (Ton /Fed) | Total (Ton /Fed) |
| Berssem | T1(100% NPK) | 8.83 | 3.90 | 2.35 | 15.08 | 9.32 | 3.84 | 2.37 | 15.53 |
|  | T2 (75%NPK+ AMF) | 9.02 | 4.33 | 2.41 | 15.76 | 9.59 | 4.13 | 2.39 | 16.11 |
|  | T3 (50%NPK+ AMF) | 8.25 | 3.86 | 2.07 | 14.18 | 8.64 | 3.73 | 2.11 | 14.48 |
|  | T4 (75%NPK+ Microbean) | 8.30 | 3.89 | 2.15 | 14.34 | 8.69 | 3.80 | 2.13 | 14.62 |
|  | T5 (50%NPK+ Microbean) | 7.81 | 3.30 | 1.96 | 13.07 | 8.16 | 3.26 | 1.98 | 13.40 |
| S. beet | T1(100% NPK) | 8.01 | 3.57 | 2.13 | 13.71 | 8.71 | 3.22 | 2.17 | 14.10 |
|  | T2 (75%NPK+ AMF) | 8.17 | 3.64 | 2.16 | 13.97 | 8.77 | 3.40 | 2.18 | 14.35 |
|  | T3 (50%NPK+ AMF) | 7.70 | 3.12 | 1.85 | 12.67 | 7.85 | 2.96 | 1.89 | 12.70 |
|  | T4 (75%NPK+ Microbean) | 7.76 | 3.20 | 1.90 | 12.86 | 7.89 | 3.03 | 1.91 | 12.83 |
|  | T5 (50%NPK+ Microbean) | 6.70 | 2.76 | 1.86 | 11.32 | 7.38 | 2.43 | 1.85 | 11.66 |
| Wheat | T1(100% NPK) | 6.78 | 2.75 | 1.99 | 11.52 | 7.23 | 2.61 | 2.02 | 11.86 |
|  | T2 (75%NPK+ AMF) | 6.81 | 2.81 | 2.03 | 11.65 | 7.29 | 2.69 | 2.05 | 12.03 |
|  | T3 (50%NPK+ AMF) | 6.70 | 2.35 | 1.74 | 10.79 | 7.04 | 2.26 | 1.77 | 11.07 |
|  | T4 (75%NPK+ Microbean) | 6.75 | 2.42 | 1.79 | 10.96 | 7.20 | 2.30 | 1.76 | 11.26 |
|  | T5 (50%NPK+ Microbean) | 6.63 | 2.17 | 1.71 | 10.51 | 6.75 | 2.04 | 1.68 | 10.47 |
|  | L.S.D.at 5% | 0.348 | 0.171 | ns | 0.302 | 0.283 | 0.159 | 0.283 | 0.294 |

L.S.D.: least significant differences at 5% of probability, ns: non-significant differences, T1: 100% mineral NPK, T2: 75% mineral NPK + Arbuscular Mycorrhiza fungi, T3: 50% mineral NPK + Arbuscular Mycorrhiza fungi, T4: 75% mineral NPK + Mycrobein, T5: 50% mineral NPK + Mycrobein.

*3.3. Quality Measurements*

The crude protein percentage (CP%) was significantly affected by preceding crops with fertilizer treatment, and the quality of forage was directly determined by the presence of crude protein contents in that forage, Table S1 and Figure 4. Maize is grown with any legume crop so that we can improve the total contents and in order to increase the amount of protein in the mixed forage because the maize straw is low in the amount of crude protein contents. This does not mean that cowpea cultivation will increase the protein content of maize straw, but rather that it will increase the protein content of the mixed forage. The data for CP% for mixed forage given in Table S1 and Figure 4 show the differences in protein content in treatments due to intercropping of cowpea with maize. The grown cowpea with maize after berseem exhibited higher protein contents with the values 9.83 and 9.82% when fertilized by 75% NPK + mycorrhiza (T2) in both seasons, respectively. The treatments where intercropping was carried out exhibit higher CP% compared with maize straw alone. This is due to higher protein percentage of legume crop. Similar results reported by Islam et al. [45]. Marchiol et al. [53] discovered that there is an increase in protein amounts by intercropping maize with legumes compared to sole maize. To increase the protein content of cereals, it is necessary to grow cereals with legumes [54]. Krishna et al. [55]. Note that there is an increase in the amount of protein content by loading maize with fodder cowpea compared to sole maize.

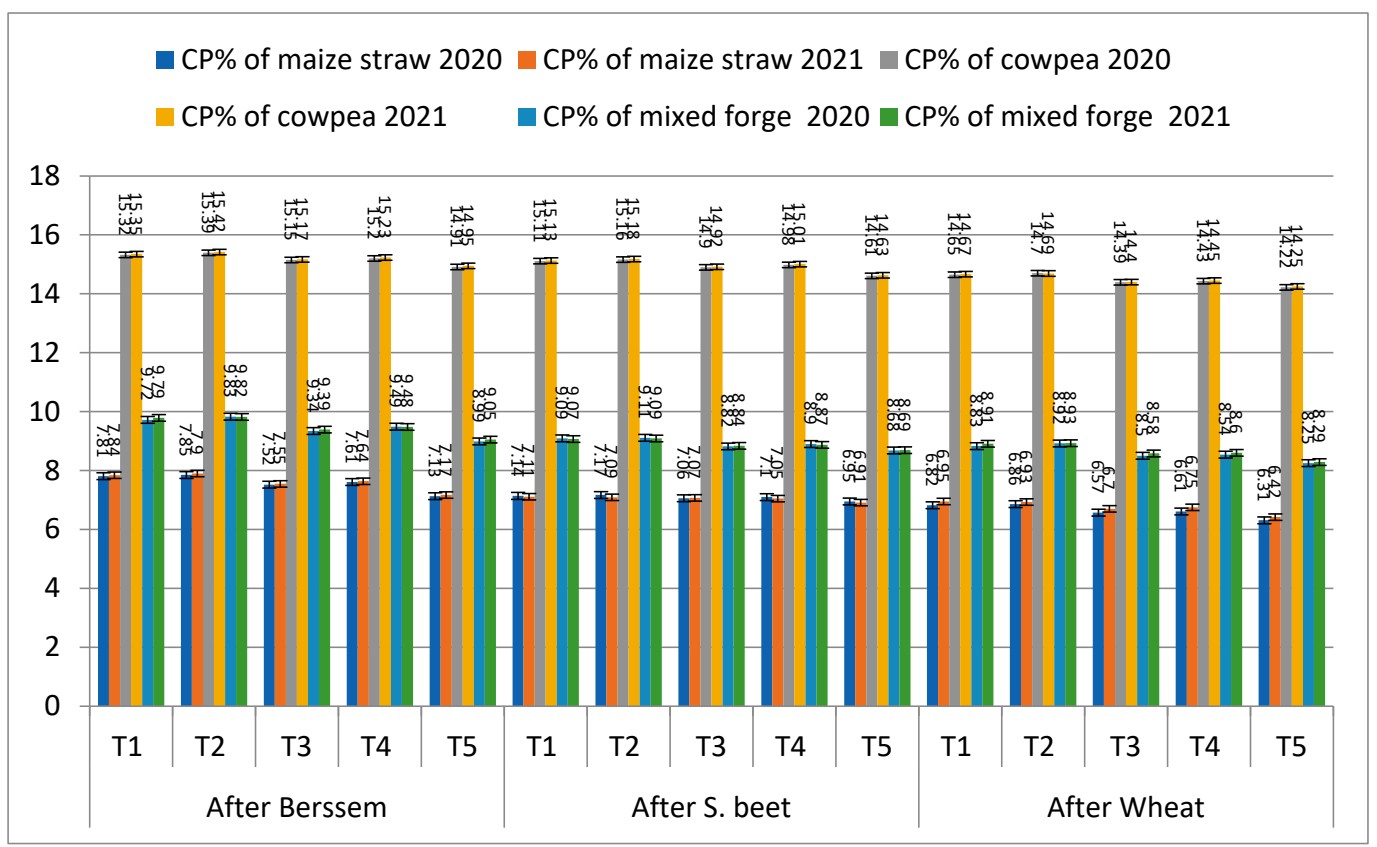

**Figure 4.** The effect of preceding crops and NPK as well as biofertilizers on the crude protein percentage of maize straw intercropping with forage cowpea and mixed during 2020 and 2021 seasons. T1: 100% mineral NPK, T2: 75% mineral NPK + Arbuscular Mycorrhiza fungi, T3: 50% mineral NPK + Arbuscular Mycorrhiza fungi, T4: 75% mineral NPK + Mycrobein, T5: 50% mineral NPK + Mycrobein.

Means of crude fiber percentage (CF%), presented in Table S2 and Figure 5, revealed that grown mixture forage after wheat by 38.04 and 37.86% were superior to grown after sugar beet by 36.28 and 35.93% in the production of significantly, compared to grown mixture forage after berseem. Nonetheless, the fertilized mixture forage by (T5) treatment resulted in the production of the significantly highest CF% by 37.68 and 37.20% compared with 35.08 and 34.74% when fertilized (T2) treatment. There is an inverse relationship between the contents of crude fiber and the quality of the forage. If the forage contains a lower amount of crude fiber, it is better quality because the high proportion of crude fiber leads to a decrease in digestibility. Where sole maize was sown exhibited higher CF% as compared to all other treatments. There is a significant decrease in the amount of CF% when cereals are grown with legumes [56]. It was observed that when sorghum was grown with cowpea, sorghum showed higher amounts of crude fiber, and only sorghum showed higher crude fiber content than that grown with cowpea [57].

The crude ash percentage (CA%), as presented in Table S2 and Figure 6. Similarly, grown mixture forage after berseem resulted in the significantly highest CA% with 10.89 and 11.14% in both seasons, respectively. Among the tested fertilizer treatments, (T2) resulted the significantly highest CA% (10.92 and 11.21%). Therefore, the planting after berseem and fertilized by (T2) resulted the highest significantly, and reached 11.82 and 12.08% in both seasons. CA% of mixed forage (maize straw + cowpea) where sole maize was sown showed minimum CA% as compared to all other treatments. Where sole cowpea shown maximum CA%; these findings are harmony in line with [52,56]. They found that when cereal crops were planted with legumes, the ash content percentage was high. These

differences in results may be due to variation in soil fertility, environmental conditions and choice of species, along with competitive relationships and yield advantages.

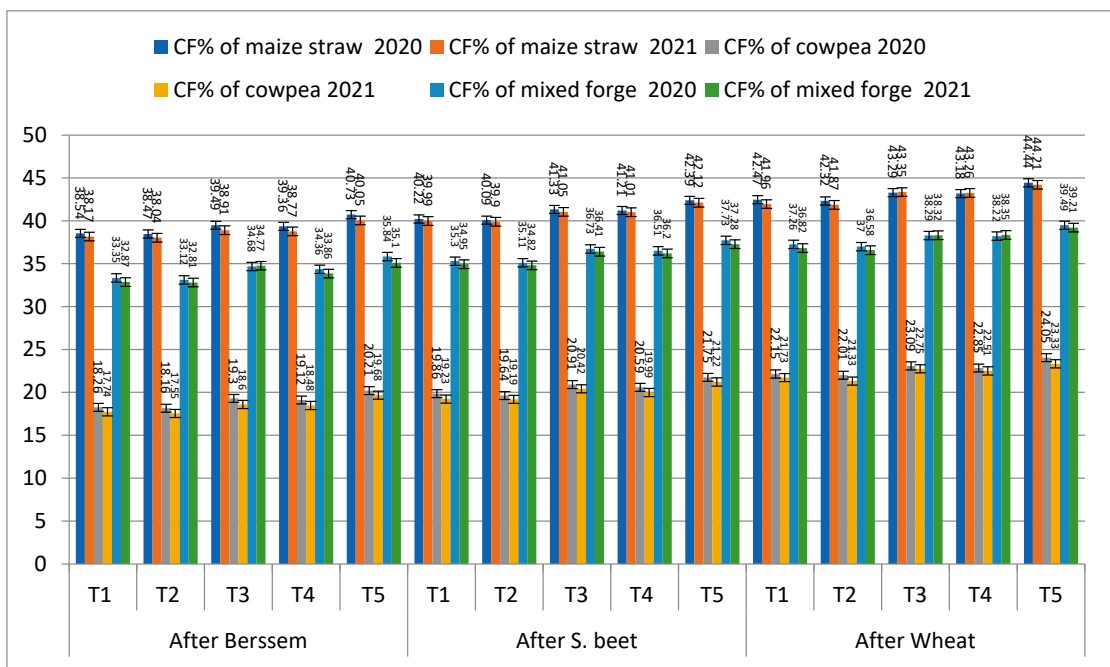

**Figure 5.** The effect of preceding crops and NPK as well as biofertilizers on the crude fiber percentage of maize straw intercropping with forage cowpea and mixed during 2020 and 2021 seasons. T1: 100% mineral NPK, T2: 75% mineral NPK + Arbuscular Mycorrhiza fungi, T3: 50% mineral NPK + Arbuscular Mycorrhiza fungi, T4: 75% mineral NPK + Mycrobein, T5: 50% mineral NPK + Mycrobein.

Increasing production per unit area by increasing the rate of land utilization for cultivation of cowpea with maize under the sequence cultivation system leads to an increase in soil fertility, compared to repeated cultivation and using the best fertilization system to reduce mineral fertilizers and reduce pollution.

3.3.1. Land Equivalent Ratio (LER)

The data revealed that the values of LER were more than 1 for all treatments when calculated based on 50% + 100% of cowpea and maize, indicating a clear land usage in the intercropping system. The data obtained in Table 9 indicate that the highest values 1.51 and 1.6 were achieved with cultivation after berseem and the application fertilizer system 75% NPK + Mycorrhiza (T2), confirming their superiority over the other treatments in uplifting the dry matter yield of the intercrops, which is the main target of the study. Therefore, the lowest values (1.2 and 1.21) were recorded with cultivation after wheat and application of 50% NPK + Mycrobein (T5) in both seasons, respectively. Salama et al. [58] reviled that even though all LER values were more than one, a clear advantage was reported for 2:2 ridges of maize and forge cowpea compared to other intercropping patterns.

Increasing production per unit area by increasing the rate of land utilization for cultivation of cowpea with maize under the sequence cultivation system leads to an increase in soil fertility, compared to repeated cultivation and using the best fertilization system to reduce mineral fertilizers and reduce pollution.

3.3.2. Land Equivalent Ratio (LER)

The data revealed that the values of LER were more than 1 for all treatments when calculated based on 50% + 100% of cowpea and maize, indicating a clear land usage in the intercropping system. The data obtained in Table 9 indicate that the highest values 1.51 and

1.6 were achieved with cultivation after berseem and the application fertilizer system 75% NPK + Mycorrhiza (T2), confirming their superiority over the other treatments in uplifting the dry matter yield of the intercrops, which is the main target of the study. Therefore, the lowest values (1.2 and 1.21) were recorded with cultivation after wheat and application of 50% NPK + Mycrobein (T5) in both seasons, respectively. Salama et al. [58] reviled that even though all LER values were more than one, a clear advantage was reported for 2:2 ridges of maize and forge cowpea compared to other intercropping patterns.

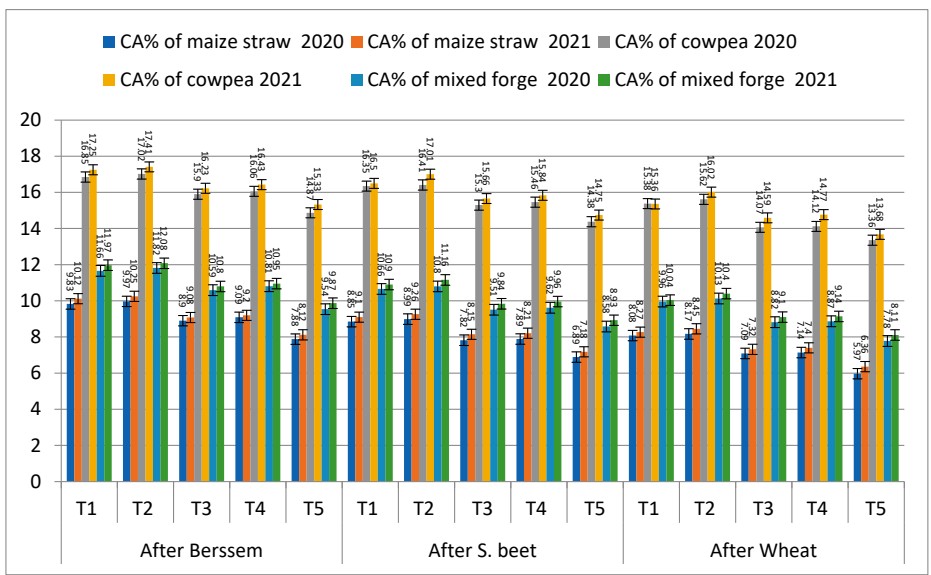

**Figure 6.** The effect of preceding crops and NPK as well as biofertilizers on the crude ash percentage of maize straw intercropping with forage cowpea and mixed during 2020 and 2021 seasons. T1: 100% mineral NPK, T2: 75% mineral NPK + Arbuscular Mycorrhiza fungi, T3: 50% mineral NPK + Arbuscular Mycorrhiza fungi, T4: 75% mineral NPK + Mycrobein, T5: 50% mineral NPK + Mycrobein.

**Table 9.** The effect of preceding crops and NPK as well as biofertilizers on land equivalent ratio (LER), relative crowding coefficient (K) and Aggressivity (A) for maize intercropping with cowpea during 2020 and 2021 seasons.

| Treatments | | Land Equivalent Ratio (LER) | | | | | | Relative Crowding Coefficient (K) | | | | | | Aggressivity (A) | | | |
|---|---|---|---|---|---|---|---|---|---|---|---|---|---|---|---|---|---|
| | | 2020 | | | 2021 | | | 2020 | | | 2021 | | | 2020 | | 2021 | |
| | | Lm | Lco | LER | Lm | Lco | LER | Km | Kco | K | Km | Kco | K | Am | Aco | Am | Aco |
| Berssem | T1 | 0.74 | 0.72 | 1.46 | 0.82 | 0.74 | 1.56 | 2.82 | 2.57 | 7.25 | 4.48 | 2.83 | 12.68 | +0.037 | −0.037 | +0.158 | −0.158 |
| | T2 | 0.76 | 0.75 | 1.51 | 0.83 | 0.77 | 1.60 | 3.11 | 3.04 | 9.45 | 4.68 | 3.28 | 15.35 | +0.009 | −0.009 | +0.115 | −0.115 |
| | T3 | 0.71 | 0.68 | 1.39 | 0.80 | 0.69 | 1.49 | 2.49 | 2.09 | 4.58 | 3.89 | 2.22 | 8.64 | +0.074 | −0.074 | +0.213 | −0.213 |
| | T4 | 0.74 | 0.69 | 1.43 | 0.81 | 0.70 | 1.51 | 2.93 | 2.16 | 6.33 | 4.39 | 2.28 | 10.01 | +0.102 | −0.102 | +0.238 | −0.238 |
| | T5 | 0.66 | 0.62 | 1.28 | 0.69 | 0.64 | 1.33 | 1.90 | 2.60 | 4.94 | 2.21 | 1.76 | 3.89 | +0.063 | −0.063 | +0.108 | −0.108 |
| S. beet | T1 | 0.75 | 0.69 | 1.44 | 0.81 | 0.68 | 1.49 | 2.63 | 2.17 | 5.71 | 4.18 | 2.15 | 8.99 | +0.121 | −0.121 | +0.250 | −0.250 |
| | T2 | 0.74 | 0.70 | 1.44 | 0.81 | 0.69 | 1.50 | 2.79 | 2.31 | 6.44 | 4.10 | 2.27 | 9.31 | +0.076 | −0.076 | +0.219 | −0.219 |
| | T3 | 0.69 | 0.63 | 1.32 | 0.74 | 0.61 | 1.35 | 2.25 | 1.72 | 3.87 | 2.79 | 1.59 | 4.44 | +0.118 | −0.118 | +0.147 | −0.147 |
| | T4 | 0.69 | 0.64 | 1.33 | 0.75 | 0.62 | 1.37 | 2.21 | 1.80 | 3.98 | 2.99 | 1.64 | 4.90 | +0.093 | −0.093 | +0.257 | −0.257 |
| | T5 | 0.64 | 0.57 | 1.21 | 0.70 | 0.57 | 1.27 | 1.77 | 1.30 | 2.30 | 2.31 | 1.29 | 2.98 | +0.145 | −0.145 | +0.267 | −0.267 |
| Wheat | T1 | 0.71 | 0.60 | 1.31 | 0.72 | 0.60 | 1.32 | 2.41 | 1.49 | 3.18 | 2.60 | 1.46 | 3.80 | +0.217 | −0.217 | +0.243 | −0.243 |
| | T2 | 0.71 | 0.61 | 1.32 | 0.72 | 0.61 | 1.33 | 2.45 | 1.53 | 3.75 | 2.60 | 1.54 | 4.00 | +0.211 | −0.211 | +0.233 | −0.233 |
| | T3 | 0.70 | 0.56 | 1.26 | 0.71 | 0.56 | 1.27 | 2.29 | 1.27 | 2.90 | 2.39 | 1.26 | 3.01 | +0.273 | −0.273 | +0.295 | −0.295 |
| | T4 | 0.70 | 0.57 | 1.27 | 0.71 | 0.57 | 1.28 | 2.37 | 1.32 | 3.13 | 2.47 | 1.31 | 3.24 | +0.269 | −0.269 | +0.290 | −0.290 |
| | T5 | 0.65 | 0.55 | 1.20 | 0.67 | 0.54 | 1.21 | 1.90 | 1.20 | 2.28 | 2.01 | 1.12 | 2.25 | +0.220 | −0.220 | +0.281 | −0.281 |
| L.S.D.at 5% | | 0.017 | ns | 0.027 | 0.028 | ns | 0.044 | 0.046 | 0.042 | 0.053 | 0.025 | 0.037 | 0.054 | − or + 0.007 | | − or + 0.009 | |

L.S.D.: least significant differences at 5% of probability, ns: non-significant differences, T1: 100% mineral NPK, T2: 75% mineral NPK + Arbuscular Mycorrhiza fungi, T3: 50% mineral NPK + Arbuscular Mycorrhiza fungi, T4: 75% mineral NPK + Mycrobein, T5: 50% mineral NPK + Mycrobein.

### 3.3.3. Relative Crowding Coefficient (RCC)

The relative crowding coefficient (RCC) was significantly affected by preceding winter crops and fertilizer mineral and biofertilizer in mixture in both seasons as shown in Table 9. The values of K were greater than unit for all treatments from 50% + 100% of cowpea and maize, indicating a clear yield advantage in the intercropping system in both seasons. The highest values of K by planting of intercropping component following a berseem, while lowest values after wheat. As for fertilization treatments, the treatment of 75% NPK + Mycorrhiza (T2) gave the values in both seasons. From here were the best yield advantage of 9.45 and 15.35 were achieved by planting intercropping component after the berseem and application of 75% NPK + Mycorrhiza (T2), while the lowest results were 2.28 and 2.25 that achieved by intercropping system after wheat with application of 50% NPK + Mycorbein (T5) in the two growing seasons, respectively. Different crops have different periods of growth and development, and therefore a single crop may provide protection from erosion forces during a period of the year, and may also improve the physical, chemical and biological properties of the soil; thus, controlling erosion and maximizing the production of the crop by maintaining soil moisture and combating diseases and pest infestations. However, another crop may not provide this same protection [59].

### 3.3.4. Aggressivity (A)

With respect to aggressivity, the third scale of competitive relations affected by preceding crops and fertilizer treatments in both seasons. It is known that an aggressivity value of zero indicates that, both component crops are equally competitive. For any other situation, both crops will have the same numerical value by positive for the dominant crop and negative for the dominated one. The results in Table 9 noticed that the component crops did not compete equally. Regardless, the intercropping pattern was apositive sign for maize and negative for cowpea, thereby the maize was dominant while cowpea was dominated of all treatments. This means that maize had higher aggressivity than cowpea under different preceding crops and fertilizer treatments in this study. The aggressivity recorded the best values of 0.009 and 00115, with minimum aggressivity by planting after berseem at (T2) fertilizer treatment, while no significance differences found between grown intercropping component after berseem when fertilized T2 and T5 treatments. While the weak values were recorded by planting intercropping component after sugar beet and wheat at (T5) fertilizer treatment, with a maximum aggressivity. Where planting intercropping component after sugar beet and wheat did not reach to the 5% level of significance with (T5) fertilizer treatment. Similar results were obtained by Asem et al. [60] and Saudy [61].

### 3.4. Economics Evaluation

The data presented in Table 10 showed that all treatments achieved positive gross return and net return compared with maize monoculture crop, except the cultivation after wheat in both seasons. The results showed that planting intercropping components after berseem and added fertilizer 75% NPK + Mycorrhiza (T2) gave the highest values for gross return and net return which were 21,053.00 and 21,933.17 L.E., as well as 15,506.00 and 16,386.17 L.E. in the first and second seasons, respectively. Whereas the lowest values were 15,644.50 and 15,887.83 L.E. as well as 10,766.50 and 11,009.83 L.E. for these characters by planting after wheat, and added 50% NPK+ Mycrobein (T5) treatment in the first and second seasons, respectively. The increases were 34.25% and 44.71% for income compared with maize monoculture crop in the first and second seasons, respectively. Similar results were obtained by Asem et al. [60].

**Table 10.** The effect of preceding crops, fertilizations and their interaction on gross return (L.E.) and net return (L.E.) for maize intercropping with cowpea during 2020 and 2021 seasons.

| Treatment | | Gross Return (L.E.) | | | | | | | | Net Return (L.E.) | |
|---|---|---|---|---|---|---|---|---|---|---|---|
| | | Cowpea | | Grain Maize | | Straw Maize | | Summation | | | |
| | | 2020 | 2021 | 2020 | 2021 | 2020 | 2021 | 2020 | 2021 | 2020 | 2021 |
| Berssem | T1 | 6786.0 | 6988.5 | 12,978.0 | 13,680.0 | 668.0 | 675.7 | 20,432.0 | 21,344.2 | 14,335.0 | 15,247.2 |
| | T2 | 7092.0 | 7249.5 | 13,284.0 | 13,986.0 | 677.0 | 697.7 | 21,053.0 | 21,933.2 | 15,506.0 | 16,386.3 |
| | T3 | 6381.0 | 6516.0 | 12,528.0 | 14,076.0 | 661.0 | 657.0 | 19,570.0 | 21,249.0 | 14,647.0 | 16,326.0 |
| | T4 | 6453.0 | 6579.0 | 12,888.0 | 14,166.0 | 656.0 | 655.0 | 19,997.0 | 21,400.0 | 14,495.0 | 15,898.0 |
| | T5 | 5881.5 | 6030.0 | 11,502.0 | 11,826.0 | 627.0 | 617.3 | 18,010.5 | 18,473.3 | 13,132.5 | 13,595.3 |
| S. beet | T1 | 6169.5 | 6345.0 | 12,798.0 | 13,554.0 | 667.0 | 673.0 | 19,634.5 | 20,572.0 | 13,537.5 | 14,475.0 |
| | T2 | 6286.5 | 6457.5 | 12,636.0 | 13,518.0 | 672.0 | 659.0 | 19,594.5 | 20,634.5 | 14,047.5 | 15,087.5 |
| | T3 | 5701.5 | 5715.0 | 11,880.0 | 12,366.0 | 635.0 | 645.0 | 18,216.5 | 18,726.0 | 13,293.5 | 13,803.0 |
| | T4 | 5787.0 | 5773.5 | 11,826.0 | 12,420.0 | 643.0 | 636.3 | 18,256.0 | 18,829.8 | 12,754.0 | 13,327.8 |
| | T5 | 5094.0 | 5247.0 | 10,962.0 | 11,736.0 | 621.0 | 611.0 | 16,677.0 | 17,594.0 | 11,799.0 | 12,716.0 |
| Wheat | T1 | 5184.0 | 5337.0 | 11,160.0 | 11,448.0 | 576.0 | 590.7 | 16,920.0 | 17,375.7 | 10,823.0 | 11,278.7 |
| | T2 | 5242.5 | 5413.5 | 11,232.0 | 11,502.0 | 571.0 | 585.0 | 17,045.5 | 17,500.5 | 11,498.5 | 11,953.5 |
| | T3 | 4855.5 | 4981.5 | 11,016.0 | 11,232.0 | 533.0 | 545.0 | 16,404.5 | 16,758.5 | 11,481.5 | 11,835.5 |
| | T4 | 4932.0 | 5067.0 | 11,124.0 | 11,340.0 | 555.0 | 568.7 | 16,611.0 | 16,975.7 | 11,109.0 | 11,473.7 |
| | T5 | 4729.5 | 4711.5 | 10,386.0 | 10,638.0 | 529.0 | 538.3 | 15,644.5 | 15,887.8 | 10,766.5 | 11,009.8 |
| L.S.D. at 5% | | 202.4 | 132.4 | ns | 417.3 | ns | ns | 480.5 | 478.3 | 565.3 | 478.3 |
| Sole maize | AB | — | — | 17,550.0 | 17,172.0 | 811.0 | 802.0 | 18,361.0 | 17,974.0 | 12,264.0 | 11,877.0 |
| | AS | — | — | 17,172.0 | 16,794.0 | 809.0 | 797.0 | 17,981.0 | 17,591.0 | 11,884.0 | 11,494.0 |
| | AW | — | — | 15,822.0 | 15,930.0 | 778.0 | 767.0 | 16,600.0 | 16,697.0 | 10,503.0 | 10,600.0 |
| | Avrage | — | — | 16,848.0 | 16,632.0 | 799.0 | 788.7 | 17,647.3 | 17,420.7 | 11,550.3 | 11,323.7 |
| Sole cowpea | AB | 9427.5 | 9459.0 | — | — | — | | 9427.5 | 9459.0 | 5747.5 | 5779.0 |
| | AS | 9009.0 | 9301.5 | — | — | — | | 9009.0 | 9301.5 | 5329.0 | 5621.5 |
| | AW | 8671.5 | 8937.0 | — | — | — | | 8671.5 | 8937.0 | 4991.5 | 5257.0 |
| | Avrage | 9036.0 | 9232.5 | — | — | — | | 9036.0 | 9232.5 | 5359.3 | 5552.5 |

## 4. Conclusions

This study concluded that the treatment 75% NPK + arbuscular mycorrhiza fungi (AMFs) (T2) gave the highest values for grain yield and forage yield of maize and cowpea that are growing under intercropping system 2:2 ridges cowpea/maize. Moreover, no significant differences were found between fertilizer treatments T1 (100% NPK mineral) and T2 (75% NPK + arbuscular mycorrhiza fungi (AMF)) combination on yield and the studied characters of maize and cowpea. This means that we can reduce the use of mineral fertilizers by 25% under this study and, thus, reduce the cost of production. The intercropping system 2:2 ridges cowpea/maize produced 70% more than of its monoculture crop of yield and forge yield/fed for maize and cowpea in both seasons, and resulted in improvement quality, increased land equivalent ratio (LER), gross return and net return.

**Supplementary Materials:** The following supporting information can be downloaded at: https://www.mdpi.com/article/10.3390/agriculture12111934/s1, Table S1: Effect of preceding crops and NPK as well as biofertilizers on the crude protein percentage of maize straw intercropping with forage cowpea and mixed during 2020 and 2021 seasons; Table S2: Effect of preceding crops and NPK as well as biofertilizers on the crude fiber percentage of maize straw intercropping with forage cowpea and mixed during 2020 and 2021 seasons; Table S3: Effect of preceding crops and NPK as well as biofertilizers on the crude ash percentage of maize straw intercropping with forage cowpea and mixed during 2020 and 2021 seasons.

**Author Contributions:** Conceptualization, A.A.M.Z.E.-D., M.H.M.K., R.A.A., S.M.A.-Q., M.M.A.A.-A. and Y.A.A.H.; data curation, A.S.M., S.M.A.-Q. and Y.A.A.H.; formal analysis, A.A.M.Z.E.-D., M.H.M.K., M.S.A., R.A.A., S.M.A.-Q., M.M.A.A.-A. and Y.A.A.H.; funding acquisition, M.S.A., R.A.A., A.S.M., N.A.A.-H. and S.M.A.-Q.; investigation, M.S.A., N.A.A.-H. and Y.A.A.H.; methodology, A.A.M.Z.E.-D., M.H.M.K., M.S.A., R.A.A., M.M.A.A.-A. and Y.A.A.H.; project administration, A.A.M.Z.E.-D. and M.M.A.A.-A.; resources, A.A.M.Z.E.-D., M.H.M.K., M.M.A.A.-A. and Y.A.A.H.; software, M.S.A., R.A.A., A.S.M. and N.A.A.-H.; supervision, M.M.A.A.-A.; validation, A.S.M., N.A.A.-H. and S.M.A.-Q.; visualiza-tion, A.S.M. and N.A.A.-H.; writing—original draft, A.A.M.Z.E.-D., M.H.M.K., M.M.A.A.-A. and Y.A.A.H.; writing—review and editing, A.A.M.Z.E.-D., M.H.M.K., M.M.A.A.-A. and Y.A.A.H. All authors have read and agreed to the published version of the manuscript.

**Funding:** This research received no external funding.

**Institutional Review Board Statement:** Not applicable.

**Data Availability Statement:** Not applicable.

**Conflicts of Interest:** The authors declare no conflict of interest.

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
