# Peer review of "Effect of Mycorrhiza Fungi, Preceding Crops, Mineral and Bio Fertilizers on Maize Intercropping with Cowpea"

_agriculture, doi:10.3390/agriculture12111934_

Round 1

Reviewer 1 Report

- Abstract section is too long. Please delete unnecessary details.

- L 21-22: Scientific names should be italic.

- For what purpose winter crops been used in this experiment?

- Please add values for mentioned traits in abstract section.

- Why was the control treatment not used in this study?

- L38: the land use efficiency index (LUE) is different with land equivalent ratio (LER). Please check it.

- L 49: Scientific names should be italic.

- L49-51: reference?

- L 57: Scientific names should be italic.

- Ofori et al. (2014) should be deleted

- L82: (Mishra, 2019 and Ayele, 2020) should be deleted

- L92: (Horvatic et al., 2018 and Yang et al., 2018) should be deleted

- L126: Gomaa et al., (2021) should be deleted

- The novelty is missed. Please highlight the novelty in introduction and discussion section.

- The hypothesis should be added.

- The objectives should be added at the end of introduction.

- Among different treatments, why not compared the intercropping data with mono-cropping?

- Among different fertilizer sources, why not compared different fertilizer sources with control (no-fertilization)?

- L227: the land use efficiency index (LUE) is different with land equivalent ratio (LER).

- L258: The statistical analysis is not clear. Please explain ‘The data obtained were analyzed according to [39]’.

- Please add the main reasons for increasing or decreasing measured traits based on the obtained results.

Fig 1: The means comparison results should be added in this figure.

- Please add standard errors in figures.

- The discussion is missed. The author firstly reported the results and after that compared the results with previously published studies. This is not a review article. For example, in the lines 272-282, the authors reported similar results published in previous studies.

- Why the results of experimental factors reported differently and again reported the interaction results?

- L312: Similar results should be deleted. Please add discussion for increasing and decreasing the measured traits.

Author Response

Author's Reply to the Review Report (Reviewer 1)

Comments and Suggestions for Authors

- Abstract section is too long. Please delete unnecessary details. Done

- L 21-22: Scientific names should be italic. Done

- For what purpose winter crops been used in this experiment?

Winter crops were used in this experiment to study their effect on the subsequent summer crop (sown after that in the same area or plot).

- Please add values for mentioned traits in abstract section. Line 39 : the values are  ratios 

- Why was the control treatment not used in this study?

We did not use treatment zero and we used treatment 100% NPK as control because we aim to study and apply the possibility of reducing the recommended fertilizer rates in the cultivation of the crops under study

- L38: the land use efficiency index (LUE) is different with land equivalent ratio (LER). Please check it. Corrected to (LER)  

- L 49: Scientific names should be italic. Done

- L49-51: reference? Done

- L 57: Scientific names should be italic. Done

- Ofori et al. (2014) should be deleted Done

- L82: (Mishra, 2019 and Ayele, 2020) should be deleted Done

- L92: (Horvatic et al., 2018 and Yang et al., 2018) should be deleted Done

- L126: Gomaa et al., (2021) should be deleted Done

- The novelty is missed. Please highlight the novelty in introduction and discussion section. Done

- The hypothesis should be added. Done

- The objectives should be added at the end of introduction. Done

- Among different treatments, why not compared the intercropping data with mono-cropping?

The data of intercropping with monoculture was compared in Table 4 and 7, but we erred in expressing the monoculture using solid maize and solid cowpea instead of monoculture and cowpea monoculture. This error has been corrected

- Among different fertilizer sources, why not compared different fertilizer sources with control (no-fertilization)?

We did not use treatment zero (no-fertilization) and we used treatment 100% NPK as control because we aim to study and apply the possibility of reducing the recommended fertilizer rates in the cultivation of the crops under study

- L227: the land use efficiency index (LUE) is different with land equivalent ratio (LER). Corrected to (LUE)  

- L258: The statistical analysis is not clear. Please explain ‘The data obtained were analyzed according to [39]’. Done, changed to :

The data obtained were analyzed by split-plot design according to Snedecor and Cochran [39]

- Please add the main reasons for increasing or decreasing measured traits based on the obtained results. Done

Fig 1: The means comparison results should be added in this figure. Done

- Please add standard errors in figures. Done

- The discussion is missed. The author firstly reported the results and after that compared the results with previously published studies. This is not a review article. For example, in the lines 272-282, the authors reported similar results published in previous studies. Done

- Why the results of experimental factors reported differently and again reported the interaction results? Done

- L312: Similar results should be deleted. Please add discussion for increasing and decreasing the measured traits. Done

Submission Date

21 October 2022

Date of this review

25 Oct 2022 17:59:32

Reviewer 2 Report

See attachment file

Author Response

Report 2

  • The aim of the paper is also confusing “determine the effect of preceding crops and inoculation with mycorrhiza and mycrobein strains, intercropping, N, P and K fertilization, as well as their interaction on minerals composition of maize and cowpea seeds”. Too many variables, and this time indicating that the experiment is to see the effect on seeds and not in plants. Corrected

  • That is why I considered the paper needs to be improved for its publication. I would also like to include other additional comments I consider relevant to improve the manuscript, that I am following presented:

Writing issues:

  • Line 55: Authors wrote the word Maize, it should be maize without initial capital letter. Done

  • Line 82, 89, 92, and so on: The names of authors cited should be outside the parenthesis and then, in the parenthesis de citation number. Please check this along the manuscript. Done

  • Line 121: “Recommended doses of chemical fertilizers”. This sentence is incomplete. Is it a sentence or should it be in parenthesis as part of the previous sentence? Done and changed to and if properly managed, bio-fertilizers the same crop can be produced using recommended doses of chemical fertilizers.

  • Line 258: According to WRITE THE NAME OF AUTHOR and then add the citation. Please check along the manuscript. Done

  • Graphs: I considered the manuscript include an excess of graphs. Some of them are not informative. I recommended to include only those that highlight relevant results. Done, deleted some figures

  • Technical issues:
  • Line 71-73: Authors concluded that “Based on this, the intercropping of maize–cowpea is very useful in countries that suffer from rising water problems, such as Egypt, as this method helps to save an important percentage of irrigation water.” I just want to indicate that the maize-cowpea intercropping was the basis of the Aztec and Maya´s agriculture in Mexican prehistoric times. It is not new but might be interested that this ancestral technique is now promoted far away as it is Egypt. Okay, Thank you for this information, but the method in Egypt is not known and is not spread as important as it is.

  • Line 121 to 125: Authors indicated that “Arbuscular mycorrhiza fungi form a mutualistic symbiosis with most crops, they take up organic carbon and provide the host plants nitrogen, phosphorous, potassium, zinc, copper, manganese and selenium [26]. Mycorrhiza biofertilizer produced healthy plants and improved seed quality [27]”. I considered it is not precise that description of the function of the arbuscular mycorrhiza. Arbuscular mycorrhizas can mobilize different nutrients to the plant and organic carbon is not the most important one (the plant mainly supplies the fungi with this element). Among them, phosphorus is the most important element mobilized, but nitrogen mobilization is not significant. Ok, changed to : Arbuscular mycorrhizas fungi get carbon from plants. Arbuscular mycorrhizas can mobilize different nutrients to the plant and the plant supplies the fungi with mainly organic carbon. Among them, phosphorus is the most important element mobilized.

  • Line 125-126: Authors indicated that “In addition, mycorrhizal fungi reduce nitrogen use and increase the available phosphorous in the soil [28]”. It is not true. Arbuscular mycorrhizas are not useful to reduce the amount of nitrogen fertilizers needed and do not increases the availability of soil phosphorus as the arbuscular mycorrhiza can only optimize the mobilization of the available soil P to the plant and cannot increase the levels of P into the soil. It is important to point out that there are different types of mycorrhizas and not all interact in the same way with its plant host. Reading general information about the function of mycorrhizas might cause those misunderstandings. Corrected to : In addition, mycorrhizal fungi increase the available phosphorous in the soil [28].

  • Methods (Design of experiment):
  • Table 1 and 2 are about soil physicochemical parameters, however the two tables use different units to present the amounts of the different elements into the soil. Please, use the same units in the two tables to understand what was the effect of the preceding crops in the soil physicochemical composition. Done

  • Authors include two types of main variables: Three preceding crops and five fertilization treatments. Within the treatments, there is one 100% chemical fertilization and four which include biofertilizers (arbuscular mycorrhizal fungi and Mycobrein) combined with chemical fertilization. As it was explained before, arbuscular mycorrhizal fungi help to mobilize effectively nutrients (specially P) to the host plant. On the other hand, Mycobrein is a biofertilizer composed by free N-fixing bacteria such as Azotobacter and some P-dissolving bacteria such as Bacillus and Pseudomonas (this information is not provided by authors and should be included in the manuscript). The way that each one of the biofertilizers act is different and not comparable. I would consider that one treatment is missing, the one with the two biofertilizers together as the function of the different biological agents could be complementary.

Additionally, authors analyze maize and cowpea as individual crops, avoiding the intercropping effect that exist when they are cropped together. It is important to mention that cowpea is a legume and legumes could fix atmospheric nitrogen too if it is associated with Rhizobium and therefore increases the inputs of nitrogen into the system. It is a mistake to analyze the effect of the different variables in each crop separately. Otherwise, crop might be considered a third main variable of the experiment and do not consider that they are intercropped.

The data of intercropping with monoculture was compared in Table 4 and 7, but we erred in expressing the monoculture using solid maize and solid cowpea instead of monoculture and cowpea monoculture. This error has been corrected

  • Results:
  • In table 4, authors present the effect of the preceding crops in maize crop production. If we see Table 2 in which soil composition after preceding crops were harvested, I cannot see significant differences in soil composition among preceding plants. According to what authors write in lines 191-194, maize and cowpea were planted after preceding plants were harvested and treatments were applied after that. Although there are differences presented in Table 4, I do not think those differences can be attributed to the preceding plant effect. The results of the experiments will be the interaction of all the variables acting together, and that is why, authors should present the interaction results as the main results of the experiment and not the results of individual variables. Done
  • As I indicated before, if maize is intercropped with cowpea, then, the crop system is an intercropping and it is not correct to evaluate the effect of treatments in each single crop (maize and cowpea).

The data and results are for intercropping and monoculture, but the authors misspelled the title and expressed the monoculture using solid maize and solid cowpea instead of monoculture and cowpea monoculture. This error has been corrected.

Conclusions

Authors initiate conclusions indicating that “Mineral fertilizer increased environmental problems”. This is not a result of this work. Conclusions might only include or refer to results of the work done. I suggest to rewrite this section. Done

Round 2

Reviewer 1 Report

Dear editor,

In the new version of manuscript, the authors revised the manuscript based on the reviewers’ comments. For improving the manuscript quality, the below comments should be corrected in the manuscript.

The standard error should be added in figure 2.

 Why the means comparison was done differently among experimental factors? The interaction effects of two factors were not significantly impacted on the studied traits?

In tables, please delete the similar letters or delete LSD values. One item is enough

In table 5, please add the treatments details at the end of table.

In table 6, please add the treatments details (T1, T2,….) at the end of table.

Figure 4, 5 and 6: The means comparison should be added in data that shown in these figures.

In Figure 4, 5 and 6, please add the treatments details (T1, T2,….) at the end of each figures.

In table 9, please add the treatments details (T1, T2,….) at the end of table.

In table 10: Reduce the number of decimal places to one number.

The citation format in text is not true.

Conclusion

- This section is repetitive and should be rewritten.

- Please make sure your conclusions' section underscores the scientific value-added of your paper, and/or the applicability of your findings/results. Highlight the novelty of your study.

Author Response

Author's Reply to the Review Report (Reviewer 1)

Review Report (Reviewer 1)

Comments and Suggestions for Authors

Dear editor,

In the new version of manuscript, the authors revised the manuscript based on the reviewers’ comments. For improving the manuscript quality, the below comments should be corrected in the manuscript.

The standard error should be added in figure 2. Done

 Why the means comparison was done differently among experimental factors? The interaction effects of two factors were not significantly impacted on the studied traits?

Done

The probabilities of the interaction effects of two factors have been added to Tables 5 and 7

In tables, please delete the similar letters or delete LSD values. One item is enough Done

In table 5, please add the treatments details at the end of table. Done

In table 6, please add the treatments details (T1, T2,….) at the end of table. Done

Figure 4, 5 and 6: The means comparison should be added in data that shown in these figures. Done

In Figure 4, 5 and 6, please add the treatments details (T1, T2,….) at the end of each figures. Done

In table 9, please add the treatments details (T1, T2,….) at the end of table. Done

In table 10: Reduce the number of decimal places to one number. Done

The citation format in text is not true. Done

Conclusion

- This section is repetitive and should be rewritten.

- Please make sure your conclusions' section underscores the scientific value-added of your paper, and/or the applicability of your findings/results. Highlight the novelty of your study. Done

Reviewer 2 Report

The paper improved with the changes introduced. I suggest to choose a better title, shortly, simple and concrete that summarize the results obtained in the paper

Author Response

Author's Reply to the Review Report (Reviewer 2)

Review Report (Reviewer 2)

Comments and Suggestions for Authors

The paper improved with the changes introduced. I suggest to choose a better title, shortly, simple and concrete that summarize the results obtained in the paper

Effect of Mycorrhiza Fungi, Preceding Crops, Mineral and Bio Fertilizers on Maize Intercropping With Cowpea
